# Burst Spinal Cord Stimulation for the Treatment of Cervical Dystonia with Intractable Pain: A Pilot Study

**DOI:** 10.3390/brainsci10110827

**Published:** 2020-11-07

**Authors:** Takeshi Shimizu, Tomoyuki Maruo, Shimpei Miura, Yuki Kimoto, Yukitaka Ushio, Satoshi Goto, Haruhiko Kishima

**Affiliations:** 1Department of Neurosurgery, Parkinson’s Disease Research Center, KKR Otemae Hospital, Osaka 540-0008, Japan; takeshi.shimizu32@gmail.com (T.S.); yushio@trad.ocn.ne.jp (Y.U.); 2Department of Neurosurgery, Kansai Rosai Hospital, Hyogo 660-8511, Japan; 3Department of Neurosurgery, Graduate School of Medicine, Osaka University, Osaka 565-0871, Japan; s-miura@nsurg.med.osaka-u.ac.jp (S.M.); yykh2yk@yahoo.co.jp (Y.K.); hkishima@nsurg.med.osaka-u.ac.jp (H.K.); 4Department of Neurodegenerative Disorders Research, Institute of Biomedical Sciences, Graduate School of Medical Sciences, Tokushima University, Tokushima 770-8503, Japan; sgoto@tokushima-u.ac.jp

**Keywords:** cervical dystonia, burst spinal cord stimulation, neuropathic pain, movement disorder

## Abstract

Pain is the most common and disabling non-motor symptom in patients with cervical dystonia. Here, we report four patients with painful cervical dystonia in whom burst spinal cord stimulation (SCS) in the cervical region produced sustained and significant improvements in both dystonic pain and motor symptoms. Further studies need to be performed to investigate the clinical efficacy of burst SCS for patients with cervical dystonia.

## 1. Introduction

Cervical dystonia (CD), the most common form of focal dystonia, is characterized by involuntary contractions of neck muscles that frequently cause repetitive twisting movement or abnormal posture [1,2]. Patients with CD often experience intractable pain in the cervical region [1,2,3]. Pain in patients with CD is considered to originate from muscles and reflects altered central processing of nociceptive stimuli caused by sustained muscle contraction and dysfunction of the neurotransmission system involving the basal ganglia nuclei [4,5]. Spinal cord stimulation (SCS) using the tonic stimulation mode has been used as an effective treatment for intractable neuropathic pain caused by a wide variety of etiologies [6]. Intriguingly, SCS has also been utilized for the treatment of movement disorders that include focal dystonia [7,8]. Based on the observation that sensory stimulation, i.e., a sensory trick, can dampen the severity of dystonic symptoms, tonic SCS of the cervical region, i.e., the C1–C2 level, was previously used for the treatment of CD [7,9]; however, its long-term effects on dystonic motor symptoms remain to be determined [8]. Burst SCS has recently emerged as an alternative intervention for pain reduction without the mandatory paresthesia [10]. A growing body of evidence indicates that in neuropathic patients, burst SCS can relieve pain while simultaneously improving functional and psychological outcomes [11]. In this single-center case series, we present the therapeutic impact of burst SCS in patients with painful CD.

## 2. Materials and Methods

### 2.1. Patients

Patients with burst SCS fulfilling the following criteria were included in the study: (1) diagnosis of CD refractory to ordinary medical treatments, (2) male or female aged 18–80 years, (3) intractable pain not caused by an etiology other than CD, and (4) written informed consent to participate in the study. The clinical characteristics of four patients with painful CD who underwent burst SCS are summarized in Table 1. Preoperative magnetic resonance imaging showed no apparent brain or spinal cord lesions in any of the patients. At the time of surgery, the mean age was 62 (range 51–75) years and the mean disease duration was 11 (range 5–20) years. Three patients, except for Patient 4, had previously received botulinum toxin (BTX) therapy, without satisfactory outcomes. No patient had received BTX for at least 3 years prior to surgery. All patients received medical treatment that included clonazepam (1.0–3.0 mg/day), trihexyphenidyl (3.0–12 mg), valproic acid (600 mg/day), tizanidine (3 mg/day), or baclofen (15 mg/day) before and after the surgery. After receiving a detailed explanation of the surgery, all patients provided written informed consent.

### 2.2. Clinical Assessments

For subjective measurements of pain severity, the visual analog scale (VAS) and the Japanese version of the Short-Form McGill Pain Questionnaire, which provides a list of Japanese words that describe some of the different qualities of pain [12], were used. For evaluating the severity of dystonic motor symptoms, we also carried out video image analysis to measure the Toronto Western Spasmodic Torticollis Rating Scale (TWSTRS) scores and the Burke–Fahn–Marsden Dystonia Rating Scale (BFMDRS) scores.

### 2.3. Surgical Procedures

The surgery was performed under local anesthesia. For the trial, two Octrode^®^ leads (Abbott Medical, Plano, TX, USA) were inserted percutaneously into the epidural space of the spine under X-ray radioscopy. The upper edges of both electrodes were finally positioned slightly lateral to the midline at the C2 level after obtaining the stimulation-induced paresthesia over the region of dystonic pain (Figure 1). During the trial period, a pulse generator was connected to the electrodes outside the body to conduct burst stimulation SCS. The stimulation parameters comprised the delivery of five spikes at 500 Hz, 40 times per second, with a pulse width of 1000 μS. The stimulation intensity was set to 50% of the sensory threshold. The combination of stimulation electrodes was changed daily to determine optimal stimulating paradigms to achieve the best pain relief. On the last day of the 7-day trial period, the percutaneous trial leads were removed. A few weeks after the trial period, the patients underwent permanent implantation of a pulse generator and two Octrode^®^ leads (Abbott Medical, Plano, TX, USA) through the same procedure used in the trial stimulation under local anesthesia. The parameters for stimulation were set to those used to achieve optimal stimulation in the trial periods. Burst SCS was continuously performed during the follow-up periods. The position of the leads was confirmed by X-ray on the day after the operation and at least once a year during the follow-up period.

### 2.4. Statistical Analysis

With respect to the four clinical assessments (TWSTRS and BFDRS), the Student’s *t*-test following Shapiro–Wilk normality test was employed to assess the statistical significance between preoperative baseline and the last follow-up. All values were expressed as means ± S.D. *p*-Values of less than 0.05 were considered to be statistically significant.

## 3. Results

All four patients with painful CD underwent burst SCS in the cervical region. There were no issues related to lead movement during the follow-up period. No perioperative complication was reported in any patients. The mean follow-up period was 21 months (range: 6–42 months). The clinical outcomes of the four patients with painful CD who underwent burst SCS are summarized in Figure 2. According to the Shapiro–Wilk test for normality, TWSTRS and BFDRS were normally distributed (*p* > 0.05 for both).

### 3.1. Relief of Dystonic Pain

Burst SCS led to a marked improvement in dystonic pain in all patients who did not report pain relapse after the pain relief by SCS. As shown in Figure 2A, the mean VAS score decreased from 83.8 ± 2.4 (range, 77–88) at baseline to 3.8 ± 3.8 (range, 0–15) at last follow-up. The mean improvement in VAS was 95.6% ± 8.8% (range, 82.4%–100%). As shown in Figure 2B, the mean SF-MPQ2 score also decreased from 73.3 ± 13.9 (range, 48–105) at baseline to 2.5 ± 2.5 (range, 0–10) at last follow-up. The mean improvement in SF-MPQ2 was 97.2% ± 5.7% (range, 88.6%–100%).

### 3.2. Relief of Dystonic Motor Symptoms

Burst SCS led to a significant improvement in dystonic motor symptoms in all patients; the dystonic postures and movements were progressively alleviated by the burst SCS. As shown in Figure 2C, the mean TWSTRS score decreased from 46.8 ± 3.6 (range, 36.5–53.0) at baseline to 23.9 ± 4.0 (range, 14.5–34.0) at last follow-up (*p* < 0.001). The mean improvement in TWSTRS was 49.9% ± 10.3% (range, 35.8%–60.3%). As shown in Figure 2D, the mean BFMDRS score also decreased from 16.3 ± 5.3 (range, 6–29) at baseline to 3.4 ± 1.3 (range, 0.5–6.0) at last follow-up (*p* < 0.05). The mean improvement in BFMDRS was 82.1% ± 7.6% (range, 73.8%–91.9%).

## 4. Discussion

Patients with CD often suffer from intractable dystonic pain [1,2,3]. The present study revealed that burst SCS achieved long-lasting, significant relief in CD-associated pain in four patients with painful CD, similar to that observed in patients with other neuropathic pain [10,11]. Although the mechanism underlying dystonia-associated pain remains unclear, it has been proposed that it is associated with not only nociceptive stimuli of muscular origin but also alterations in the central nervous system processing [4,5]. Through large, myelinated Aβ afferents whose collaterals ascend in the dorsal column, SCS can act on the reticular system that modulates the activities of deep brain nuclei [13,14,15]. In particular, by mimicking the brain activity, burst SCS can induce dynamic remote effects on cortical cells through “burst” firing of thalamic relay cells [16,17,18]. This evidence suggests that burst SCS exerts a therapeutic effect on dystonia-associated pain via both peripheral and central mechanisms.

This open-label study is the first to show that burst SCS can produce sustained and significant improvement in the motor symptoms of patients with CD during a long mean follow-up period of 21 months (range, 6–42 months). Since the first study by Gildenberg, who reported good results with tonic SCS utilizing high frequencies (800–1100 Hz) at the C1–C2 level for the treatment of CD [9], several studies reported the therapeutic efficacy of this approach on CD and other types of dystonia [19,20,21,22]. Dieckmann et al. also reported that continuous 1100–Hz stimulation at the C2–C4 level resulted in good outcomes at 8–12 months follow-up in patients with CD [19]. Waltz et al. reported a large series of 129 dystonic patients treated with cervical SCS. In 66 CD patients, 77% demonstrated some improvement, including 38% with marked improvement in dystonic symptoms [7]. Conversely, Fahn reported negative results, with only one patient experiencing long-lasting benefit among 25 patients with dystonia, and questioned the clinical efficacy of cervical SCS [20]. Furthermore, Broseta et al. reported poor outcomes after a mean follow-up of 41.4 months in patients treated with SCS utilizing 200–1400 Hz at the level of C2 [21]. Following these controversial results, Goetz et al. conducted a double-blind crossover trial of cervical SCS for dystonic patients and reported that dystonic symptoms were not significantly improved [22]. Based on the currently available data, it is likely that tonic SCS might provide significant short-term, but not long-term, relief of motor symptoms in patients with CD [8]. The placebo effect of mechanical sensation has been suggested as a reason for the negative outcomes following tonic SCS [8,23,24]. Therefore, we propose that burst SCS is useful to distinguish a responding patient without the placebo effect because burst SCS is known not to cause stimulation-induced paresthesia [10].

SCS is generally accepted to have the ability to alleviate the motor symptoms of CD by interrupting the tonic neck reflex pattern that is regulated by propriospinal fibers [9]. Conversely, SCS might also exert an ascending remote effect to modulate the neuronal activities of the brainstem and forebrain [25,26,27,28], raising the possibility that burst SCS also exerts a therapeutic effect on the motor symptoms of CD by normalizing the basal ganglia-thalamo-cortical circuit activity, the dysfunction of which is tightly linked to the development of focal dystonia [29]. Future studies with large cohorts should elucidate the mechanisms by which burst SCS alleviates dystonic motor symptoms.

## 5. Conclusions

Dystonic neck pain is a disabling condition that markedly impairs the quality of life in patients with CD. However, optimal therapies for painful CD are currently lacking. Among the variety of therapeutic interventions, intramuscular BTX injection is considered the first-line therapy for CD; however, few reports have addressed its efficacy and safety in patients with CD. Indeed, BTX therapy did not yield satisfactory results in any of the three patients who received the treatment in the present study. In contrast, we showed that burst SCS could produce sustained and significant improvements in dystonic pain as well as motor symptoms in patients with painful CD. Further studies are necessary to investigate the clinical efficacy of burst SCS in patients with CD.

## Figures and Tables

**Figure 1 brainsci-10-00827-f001:**
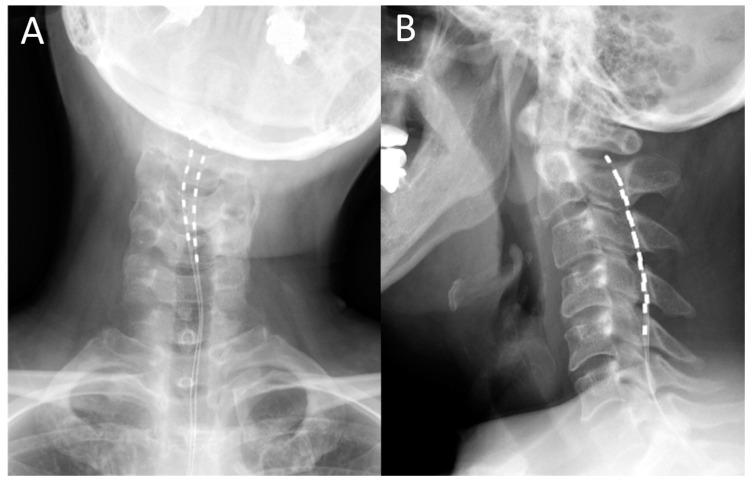
Postoperative cervical X-ray images following spinal cord stimulation. (**A**) Anteroposterior and (**B**) lateral view.

**Figure 2 brainsci-10-00827-f002:**
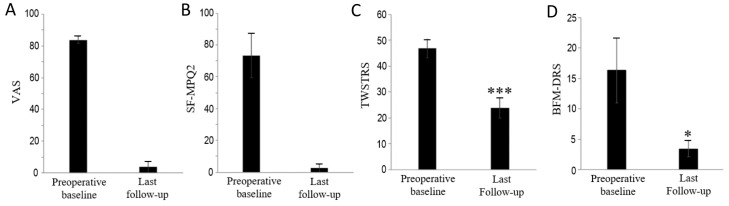
Clinical outcomes of patients with painful cervical dystonia who underwent cervical burst spinal cord stimulation. (**A**) Visual analog scale (VAS), (**B**) Short-Form McGill Pain Questionnaire (SF-MPQ-2), (**C**) Toronto Western Spasmodic Torticollis Rating Scale (TWSTRS), and (**D**) Burke–Fahn–Marsden Dystonia Rating Scale (BFMDRS).

**Table 1 brainsci-10-00827-t001:** Clinical summary of patients with painful cervical dystonia who underwent burst spinal cord stimulation.

	Patient			
	1	2	3	4
Age (year)/sex	51/F	51/F	71/F	75/M
Duration of disease (year)	8	5	20	5
Follow-up after surgery (months)	42	24	6	12
VAS scores				
Preop	85	88	77	85
Last follow-up	0	0	0	15
Percent improvement (%)	100	100	100	82.4
SF-MPQ2 scores				
Preop	105	48	52	88
Last follow-up	0	0	0	10
Percent improvement (%)	100	100	100	88.6
WSTRS scores				
Preop	49	48.5	53	36.5
Last follow-up	23	24	34	14.5
Percent improvement (%)	53.1	50.5	35.8	60.3
BFMDRS motor scores				
Preop	9	6	29	21
Last follow-up	1.5	0.5	6	5.5
Percent improvement (%)	83.3	91.7	79.3	73.8

Abbreviations: F, female; M, male; VAS, visual analog scale; SF-MPQ2, Short-Form McGill Pain Questionnaire; TWSTRS, Toronto Western Spasmodic Torticollis Rating Scale; BFMDRS, Burke–Fahn–Marsden Dystonia Rating Scale.

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
