# Peer review of "Burst Spinal Cord Stimulation for the Treatment of Cervical Dystonia with Intractable Pain: A Pilot Study"

_brainsci, 2020, doi:10.3390/brainsci10110827_

Round 1

Reviewer 1 Report

Comments and Suggestions for Authors

Methods section requires extensive editing and inclusion of more information.

  1. It is not clear whether Botulinum toxin was administered to patients preoperatively and if it did, what evidence is there that it is not the compounding effect of SCS and botulinum toxin that produced positive effects.
  2. Please include more information about the stimulation device and the stimulation protocol. What were the stimulation parameters, continuous, few hours per day? How long was stimulation performed for? What was the reason for the trial stimulation first followed by the implantation of a generator? First it is mentioned that the trial electrodes were implanted epidurally and then that trial leads were removed from percutaneous space, please clarify in the text. How was SCS achieved during the trial phase? Please include more detail about the parameters of stimulation. Was the generator implanted during the trial as well? Were the leads explanted at any point?
  3. Please add information about whether the trial leads were inserted under anesthesia and if so, how was parenthesis achieved?

General comments:

  1. Have authors considered testing for any signs of change in cognitive function (positive or negative) in response to stimulation protocol?
  2. Were there any issues with movement of the leads and have patients been re-examined under x-ray for position of the leads in the end of the study?

Reviewer 2 Report

Comments and Suggestions for Authors

The authors present an interesting study concerning the use of spinal cord stimulation (SCS) as a treatment for cervical dystonia (CD), specifically SCS with burst stimulation. 4 patients participated in the pilot trial, all of whom experienced substantial reduction of symptoms.

I do find this pilot study well performed and important, since alternative treatment strategies, such as deep brain stimulation (DBS) may appear more invasive. I do, however, believe that the manuscript would benefit from some adjustments.

The authors should more clearly bring up previous studies utilizing SCS for CD. They reference some, but do not go into detail as to how SCS was performed in these studies nor do they clarify the results. E.g. Gildenberg (ref 9) reported quite good results with SCS utilizing high frequencies (e.g. 1.100 Hz) not producing paresthesias. Other articles provide information on similar results with similar setups.

Could the authors please provide detailed information on the products used at final implantation and not only information on products used fro trial procedure?

I do not consider using T test for non-parametric data (VAS, TWSTRS, BFMDRS, etc.) a good practice. I suggest that the authors recalculate using statistical methods designed for non-parametric data.

Could the authors please provide more insight into why patient 4 (reporting good results from botulinum toxin treatment) was considered for this treatment?

The main conclusion, in the Abstract as well as in the main text, must be reappraised. In my mind a pilot study cannot form the basis for suggestions such as "patients with painful cervical dystonia could be good candidates for positive outcome with burst SCS". For any new treatment to be properly assessed a pilot study is a good basis for larger studies (preferably RCTs) but not for treatment recommendations. The conclusion should be a call for the scientific community to provide such studies (possibly under the auspices of the authors).

there are some language errors that need to be addressed. Could the authors carefully recheck the English language?
